# The Impact of Morbid Obesity on the Health Outcomes of Hospital Inpatients: An Observational Study

**DOI:** 10.3390/jcm10194382

**Published:** 2021-09-25

**Authors:** Kellie Fusco, Campbell Thompson, Richard Woodman, Chris Horwood, Paul Hakendorf, Yogesh Sharma

**Affiliations:** 1Discipline of Medicine, The University of Adelaide, Adelaide 5005, Australia; kellie.fusco@adelaide.edu.au (K.F.); Campbell.Thompson@adelaide.edu.au (C.T.); 2College of Medicine and Public Health, Flinders University, Adelaide 5042, Australia; Richard.Woodman@flinders.edu.au; 3Department of Clinical Epidemiology, Flinders Medical Centre, Adelaide 5042, Australia; Chris.Horwood@sa.gov.au (C.H.); Paul.Hakendorf@sa.gov.au (P.H.); 4Department of General Medicine, Division of Medicine, Cardiac & Critical Care, Flinders Medical Centre, Adelaide 5042, Australia

**Keywords:** morbid obesity, health outcomes, in-hospital mortality, length of hospital stay, readmissions

## Abstract

Morbid obesity poses a significant burden on the health-care system. This study determined whether morbid obesity leads to worse health-outcomes in hospitalised patients. This retrospective-study examined nutritional data of all inpatients aged 18–79 years, with a body-mass-index (BMI) ≥ 18.5 kg/m^2^ admitted over a period of 4 years at two major hospitals in Australia. Patients were divided into 3 groups for comparison: normal/overweight (BMI 18.5–29.9 kg/m^2^), obese (BMI 30–39.9 kg/m^2^) and morbidly-obese (BMI ≥ 40 kg/m^2^). Outcome measures included length-of-hospital-stay (LOS), in-hospital mortality, and 30-day readmissions. Multilevel-mixed-effects regression was used to compare clinical outcomes between the groups after adjustment for potential confounders. Of 16,579 patients, 1004 (6.1%) were classified as morbidly-obese. Morbidly-obese patients had a significantly longer median (IQR) LOS than normal/overweight patients (5 (2, 12) vs. 5 (2, 11) days, *p* value = 0.012) and obese-patients (5 (2, 12) vs. 5 (2, 10) days, *p* value = 0.036). After adjusted-analysis, morbidly-obese patients had a higher incidence of a longer LOS than normal/overweight patients (IRR 1.04; 95% CI 1.02–1.07; *p* value < 0.001) and obese-patients (IRR 1.13; 95% CI 1.11–1.16; *p* value < 0.001). Other clinical outcomes were similar between the different groups. Morbid obesity leads to a longer LOS in hospitalised patients but does not adversely affect other clinical outcomes.

## 1. Introduction

The prevalence of overweight and obesity has been increasing worldwide over the past 40 years [1]. It is estimated that in Australia, almost two thirds of adults are either overweight or obese [2]. Approximately 1 million adults in Australia have clinically severe obesity (defined as a body mass index (BMI) ≥ 40 kg/m^2^ or ≥35 kg/m^2^ with at least one complication) [3]. A significant burden of disease in the population at large is caused by overweight (and obesity) induced comorbidity [4]. Overweight and obesity have been linked to a number of diseases including diabetes, asthma, osteoarthritis, gout, cardiovascular disease, and certain types of cancers [2,5,6]. Obesity has long been considered as a major risk factor for mortality in both the general population and in hospitalised patients. The costs of obesity and overweight to the health care system are complex to calculate [7]. There is a burden of chronic disease reflected in higher costs for care of obese within the community but there is also a burden of acute and chronic disease reflected in higher costs for the care of obese individuals within hospitals such as may be seen due to longer and more frequent intensive care unit (ICU) admissions, longer lengths of hospital stay (LOS), and higher likelihood of readmission within 30 days [8,9].

Given the association of obesity with various other co-morbidities, it might be safe to assume that overweight and obese patients will have poorer health outcomes than those with a healthy body weight. However, recent evidence [10] suggests that older individuals with several comorbidities and an elevated BMI, may have a lower mortality when compared to patients with normal BMI. This mortality benefit of obesity has been termed as the ‘obesity paradox’ and has been observed in patients with diabetes, cardiovascular disease, chronic obstructive pulmonary disease (COPD) and cancer [11,12,13,14,15]. However, other studies have not confirmed the benefits of increasing BMI on clinical outcomes [16,17].

The inconsistent evidence regarding obese patients’ clinical outcomes may be attributable to a number of factors including different cut-off values of BMI used to classify obesity in different studies [18]. In addition, studies may have methodological differences such as the source of BMI data (self-reported or objective evidence), patient selection bias and the presence of residual confounding [19,20]. In addition, very few studies have examined clinical outcomes in very severely obese patients, i.e., BMI ≥ 40kg/m^2^, especially in hospitalised patients. A major limitation of the existing studies is the lack of detailed data on associated co-morbidities, socio-economic status of the patients and ethnicity, all of which have been identified as factors which can contribute to both obesity and clinical outcomes [21,22,23].

This study was therefore designed to examine the association between morbid obesity (BMI ≥ 40kg/m^2^) and clinical outcomes in hospitalised patients after including important confounders such as the comorbidity burden, socioeconomic status and the ethnicity of the participants. The hypothesis for this study was that morbidly obese hospitalised patients will have worse clinical outcomes than non-morbidly obese patients. The primary outcome measure for this study was to determine whether morbid obesity was associated with a longer LOS. The secondary outcomes were to determine any differences in in-hospital mortality, number of complications during admission, ICU length of stay and 30-day readmissions between morbidly obese and non-morbidly obese patients.

## 2. Materials and Methods

Non-identifiable, routinely collected data from two of South Australia’s tertiary-level teaching hospitals (Flinders Medical Centre (FMC) and Royal Adelaide Hospital (RAH)) were examined. These two hospitals in South Australia have adopted a nutritional screening tool (Malnutrition Universal Screening Tool (eMUST)) [24] which has allowed for the determination of each inpatient’s BMI. These data, when linked to hospital inpatient databases, were used to determine the differences in outcomes for inpatients at these two hospitals using specific BMI groupings. The eMUST database was accessed and combined with data from the hospital inpatient database. The eMUST database includes information about the following variables: patient identification number, date of birth, date of hospital admission, date of performance of the MUST, height, weight, BMI, history of recent weight loss, impact of acute illness on nutritional status and the total MUST score. This study included information from all inpatients at FMC and RAH, who had an eMUST recorded between 1 January 2015 and 31 December 2018. The hospital inpatient dataset variables included: age, sex, Charlson comorbidity index, socio-economic category as determined by the Index of Relative Social Disadvantage (IRSD), indigenous status, LOS, admission to ICU, LOS in ICU, complications during index admission, readmissions within 7 and 30 days of discharge.

Patients with a BMI of 18.5–24.9 kg/m^2^ were classified as normal, 25–29.9 kg/m^2^ as overweight, 30–34.9 kg/m^2^ as class 1 obesity, 35–39.9 kg/m^2^ as class 2 obesity and ≥40 kg/m^2^ as class 3 obesity or morbidly obese. For this study, patients were divided into 3 groups for comparison: normal/overweight (BMI 18.5–29.9 kg/m^2^), obese (BMI 30–39.9 kg/m^2^) and morbidly obese (BMI ≥ 40 kg/m^2^). The ethical approval for this study was granted by the Southern Adelaide Human Clinical Research Ethics Committee (SAHREC) no 387.11.

### Statistics

Data were inspected for normality using histograms and differences between different groups were assessed using the one-way analysis of variance (ANOVA) or the Kruskal Wallis H test. Differences in categorical variables were analysed using the chi-squared statistics. We used the Dunn test for multiple comparisons and family wise error was adjusted by use of the Bonferroni correction. The relationships between morbid obesity and the outcome variables were examined using multilevel mixed-effect models with hospital outcomes as a random effect [25]. We used mixed-effects Poisson regression for LOS and incidence risk ratios (IRR) were determined, while in-hospital mortality, 7 and 30-day readmission rate were assessed by use of mixed-effects logistic regression models and odds ratios (OR) were determined. Patient-level factors, which have been identified as important in previous studies as potential confounders [23,26], were considered in the multilevel analysis as fixed effects. These included: age, sex, Charlson index, socio-economic status as determined by the IRSD, and indigenous status of the patients. An intercept-only model was also assessed in order to determine the intraclass correlation coefficient, which describes the proportion of the total variance that is attributable to clustering within-hospitals [27]. The appropriateness of the final model was confirmed by examining the distribution of the level 1 and level 2 residuals to check for normality. All statistical analyses were performed by of STATA version 17.0 (StataCorp LP, College Station, TX, USA). All outcomes were assessed using a 2-sided Type 1 error rate of alpha = 0.05.

## 3. Results

### 3.1. Data

The data included details of 28,907 inpatients who had an eMUST performed during hospital admission. After removal of the second and subsequent visits for any patient (*n* = 1631), underweight patients (BMI < 18.5 kg/m^2^) (*n* = 1668), and those over the age of 80 years (*n* = 9029), 16,579 patients were available for inclusion in this study (Figure 1). When characterised according to the BMI, 34.6% of these patients had a normal BMI, 31.9% were overweight, 18.9% were in obese class 1, 8.5% in obese class 2 and 6.1% were morbidly obese (Table 1). For this study, we compared the clinical characteristics and outcomes between 3 groups: normal/overweight patients *n* = 11,028 (66.5%), obese patients (obesity class 1 and 2) *n* = 4547 (27.4%) and the morbidly obese group *n* = 1004 (6.1%).

### 3.2. Characteristics

Table 2 shows baseline characteristics of morbidly obese patients when compared to the other groups. Morbidly obese patients were more likely to be younger and female, with a higher comorbidity burden as reflected by the higher Charlson index, and they belonged to a lower socio-economic status as reflected by a lower IRSD score than those who were who were normal/overweight or those who were obese (*p* < 0.05). There was no significant difference in the number of patients who identified as indigenous or Torres Strait Islanders, who belonged to the morbidly obese category when compared to the other groups (*p* < 0.05).

### 3.3. Univariate Analysis

The median (IQR) overall hospital LOS was 5 (1, 9) days. Morbidly obese patients had a significantly longer median (IQR) LOS when compared to both normal/overweight patients (5 days (2, 12) vs. 5 days (2, 11), *p* = 0.012) and obese patients (5 days (2, 12) vs. 5 days (2, 10), *p* value = 0.036) (Table 3). The overall in-hospital mortality rate was 0.88% and there was no difference in mortality in different groups (0.51% vs. 0.93% vs. 0.90, *p* = 0.439), in morbidly obese, obese and normal/overweight patients, respectively (Table 3). Other outcomes including ICU admission, duration of ICU stay, number of complications during admission and readmission within 30-days of hospital discharge were similar between different groups of patients (Table 3).

### 3.4. Adjusted Analysis

#### 3.4.1. Comparison of Outcomes between Morbidly Obese and Normal/Overweight Patients

Table 4 and Table 5 shows outcomes after multilevel mixed-effect regression models adjusted for various confounders in morbidly obese when compared to normal/overweight patients and obese patients, respectively. The proportion of total variance accounted for the two different hospitals as reflected by the intraclass correlation coefficient was small (0.001, SE 0.007, 95% CI 8.99–0.02). When compared to normal/overweight patients, the overall hospital LOS was significantly longer in morbidly obese patients (IRR 1.04; 95% CI 1.02–1.07; *p* value < 0.001) (Table 4). There was no significant difference in in-hospital mortality in morbidly obese patients when compared to the normal/overweight patients (aOR 0.65; 95% CI 0.26–1.62; *p* = 0.354) (Table 4). Similarly, other clinical outcomes such as ICU admission, length of ICU stay, complications and 30-day readmissions were also not significantly different between the two groups (Table 4).

#### 3.4.2. Comparison of Outcomes between Morbidly Obese and Obese Patients

When compared to obese patients, the overall hospital LOS was significantly longer in morbidly obese patients (IRR 1.13; 95% CI 1.11–1.16; *p* value < 0.001) **(**Table 5). Morbidly obese patients also had a significantly higher number of complications during admission when compared to obese patients (IRR 1.09; 95% CI 1.02–1.18, *p* value 0.017) (Table 5). However, there was no significant difference in in-hospital mortality in morbidly obese patients when compared to the obese group (aOR 0.66; 95% CI 0.26–1.62; *p* = 0.354) (Table 5). Other clinical outcomes such as ICU admission, length of ICU stay and 30-day readmissions were also not significantly different between the two groups (Table 5).

## 4. Discussion

The results of this study indicate that overall 6.1% of patients hospitalised during the study period were morbidly obese and these morbidly obese patients were more likely to be younger and female, with a higher comorbidity burden, and belonged to a lower socio-economic category than the non-morbidly obese group. When compared to normal/overweight patients and obese patients, morbidly obese patients had a significantly longer LOS. The number of complications were significantly higher in morbidly obese patients when compared to obese patients but not when compared to normal/overweight patients and other clinical outcomes were similar between the different groups.

The prevalence of morbid obesity according to this study was 6.1%, which is much lower than a United States (US) study [20] which included 800,417 hospitalised patients and found that 23.4% were morbidly obese. This discrepancy in the prevalence rates of morbid obesity in hospitalised patients between the two studies could partly be a reflection of in general a higher prevalence of obesity in US compared to Australia [28]. In addition, the discrepancy could be related to the differences in selection criteria between the two studies because the American study included only patients who were over the age of 40 years who were admitted with a diagnosis of cancer, chronic obstructive pulmonary disease (COPD), asthma and cardiovascular disease (CVD) compared to our inclusion of relatively younger patients presenting with a broader range of clinical diagnoses in our study. The prevalence rate of morbid obesity in our study, however, matches another study which included >18,000 patients discharged with a diagnosis of pneumonia and found that 4% of these patients were morbidly obese [29].

Our study found that morbidly obese patients have a longer LOS than the non-morbidly obese group. These results are similar to a study by Harris et al. [30], which explored 31730 diabetic foot infection/ulcer related hospitalisations and found that morbidly obese patients (BMI ≥ 40 kg/m^2^) were more likely to be younger females, mean age (56.9 years) with a higher Charlson comorbidity score than non-obese patients and stayed for a longer time in hospital (mean LOS 5.5 days vs. 5.0 days, *p* < 0.01) (adjusted mean difference 0.47; 95% CI 0.13–0.81) than the non-morbidly obese group. Their study also found that morbidly obese patients had greater hospital resource utilisation and incurred higher hospitalisation charges than non-morbidly obese patients. Similarly, a Canadian study [31] which included 7560 patients, who underwent coronary artery bypass grafting, found that morbid obesity was significantly associated with a prolonged LOS when compared to those who had normal BMI (median (IQR) LOS 7 (5, 10) days vs. 6 (5, 9) days) and speculated that this could due to an increased risk of post-operative infection in morbid obese subjects. This study, however, did not find any significant differences between pulmonary or renal complications which could have accounted for an increased LOS between morbidly obese patients and those with a normal BMI.

We found no association between morbid obesity and in-hospital mortality in our study population and the probability of making a Type 2 error (false rejection of null hypothesis when it is true) is unlikely because of the presence of reasonable sample size in our data set. In addition, we tested this association by splitting the data over individual years and found no significant association between morbid obesity and in-hospital mortality. The results of our study in terms of mortality are similar to a study by Harris et al. [30], who included 4334 morbidly obese diabetic patients admitted with foot ulcers/infections and found no significant differences in in-hospital mortality (aOR 3.89, 95% CI 0.79–19.30, *p* = 0.09) when compared to the non-morbidly obese group. Akinyemiju et al. [20] in their study involving 800,417 US patients, hospitalised with a primary diagnosis of cancer, COPD, asthma and cardiovascular disease, found that there was a bell shaped relationship between in-hospital mortality and BMI with a trend towards reduction at higher ranges of BMI. This study, however, did not specifically compare clinical outcomes between morbidly obese patients and those with normal BMI or between those in the other obese categories.

Our study results also correspond to another study by King et al. [29], who included 18,746 hospitalised patients with a discharge diagnosis of pneumonia and found that 4% of patients were morbidly obese (BMI ≥ 40 kg/m^2^) and, when compared to patients with normal weight, morbid obesity was not associated with an increased 90-day mortality (aOR 0.96, 95% CI 0.72–1.28). Similar to our study, this study also did not find any association between morbid obesity and the risk of ICU admission.

Obese individuals are at an increased risk of obstructive sleep apnoea and obesity hypoventilation syndrome and have altered lung mechanics including reduced lung volumes, decreased respiratory compliance and impaired respiratory muscle function with a reduction in gas exchange [32,33,34]. In addition, obesity has been associated with increased susceptibility to various infections including bacteraemia, surgical site infections and poor wound healing, and obese subjects demonstrate a more severe illness with certain viral infections such as influenza and COVID-19 [35,36,37,38,39]. Given this demonstrated increased risk of infection and alteration in lung function, it is logical to believe that obesity will be associated with an increased risk of pneumonia and other nosocomial infections with a resultant increased risk of ICU care and higher mortality. However, current evidence does not support this hypothesis. On the contrary, more recent data support an association between obesity and improved hospital outcomes measured in terms of mortality and complications [29,40,41]. Studies in hospitalised patients have suggested mortality benefits of obesity in pneumonia; benefits which were maintained after adjustment for age and the presence of diabetes [29,41]. The protective effects of obesity could be related to an attenuated inflammatory response to an acute infection or stress because of the fact that obesity is already associated with a low-level inflammation at baseline [42] and studies suggest that the relative increase in inflammation in response to an acute stressor in subjects with obesity may be relatively less than that seen in subjects with a normal BMI [29,43]. This could potentially lead to less tissue injury in obese subjects and thus a lesser severity of nosocomial infections with a reduction in adverse clinical outcomes.

Another possible explanation for the protective effects of obesity could be related to differences in immunological function. Obese individuals demonstrate increased leptin levels, a hormone produced by adipocytes that participates in both adaptive and innate immune responses [44,45]. Leptin promotes macrophage activation, neutrophil chemotaxis, natural killer cells cytotoxicity and lymphopoiesis, and thus, leptin helps to increase bacterial clearance [46,47]. Immunological and inflammatory effects of obesity are fascinating areas that may be playing a protective role in these patients and both deserve further research to further clarify their clinical impact.

### Limitations

The dataset contains only about 10% of patients admitted to the two institutions over the period of study which could have introduced a selection bias in this study. This poor capture of patient anthropometry is disappointing but not unusual. However, the BMI distribution in our study matches another recent Australian study [48], which captured data of all hospitalised patients. The retrospective design of this study is another limitation because of possibility of unknown confounders which could have influenced our results.

## 5. Conclusions

About 6% of patients presenting to hospitals were morbidly obese (BMI ≥ 40 kg/m^2^). Although morbid obesity was associated with a slightly longer LOS (less than half a day on average), other clinical outcomes were similar among hospitalised adult patients < 80 years of age.

## Figures and Tables

**Figure 1 jcm-10-04382-f001:**
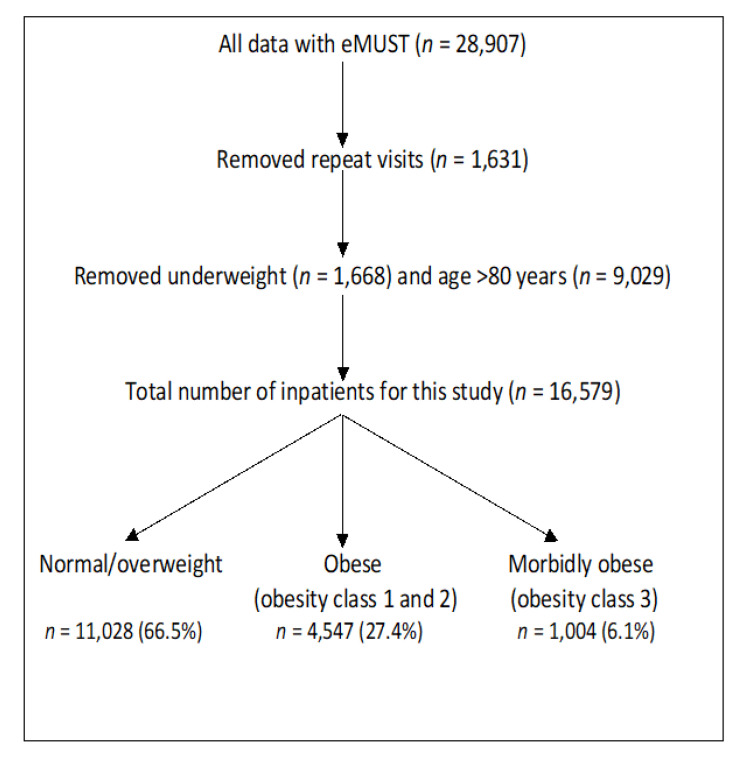
Study flow diagram.

**Table 1 jcm-10-04382-t001:** BMI categories of patients between 18–79 years excluding underweight patients.

BMI (kg/m^2^)	Normal BMI BMI 18.5–24.9	Overweight BMI 25–29.9	Obese Class 1 BMI 30–34.9	Obese Class 2 BMI 35–35.9	Obese Class 3 BMI > 40
*n* = 16,579	5731	5297	3136	1411	1004
%	34.6	31.9	18.9	8.5	6.1

BMI, body mass index.

**Table 2 jcm-10-04382-t002:** Baseline patient characteristics.

Variable	Normal/Overweight	Obese (Obesity Class 1 and 2)	Morbidly Obese (Obesity Class III)	*p* Value
Number of subjects (%)	*n* = 11,028 (66.5)	*n* = 4547 (27.4)	*n* = 1004 (6.1)	
Characteristics				
Age in years med (IQR)	60 (43,71)	61 (49,70)	56 (45,67)	<0.0001
Age categories (%)				
18–29 30–39 40–49 50–59 60–69 70–79	1346 (12.2) 1030 (9.4) 1270 (11.5) 1783 (16.2) 2397 (21.7) 3202 (29.0)	272 (5.9) 383 (8.4) 553 (12.2) 901 (19.8) 1167 (25.8) 1271 (27.9)	71 (7.1) 81 (8.1) 193 (19.2) 230 (22.9) 255 (25.4) 174 (17.3)	<0.001
Charlson Index med (IQR)	0 (0,2)	1 (0,2)	1 (0,2)	<0.0001
Gender female *n* (%)	4455 (40.4)	2073 (45.6)	629 (62.7)	<0.001
Indigenous/Torres islander status *n* (%)	328 (2.9)	134 (2.9)	43 (4.3)	0.176
IRSD Quintile *n* (%)				
Q1 (most disadvantaged) Q2 Q3 Q4 Q5 (least disadvantaged)	2505 (23.1) 2391 (22.2) 2147 (19.8) 1844 (17.0) 1942 (17.9)	1157 (25.8) 1105 (24.7) 866 (19.3) 717 (16.0) 634 (14.2)	297 (30.0) 259 (26.3) 183 (18.5) 128 (12.9) 122 (12.3)	<0.001

IQR, interquartile range; IRSD, index of relative socio-economic disadvantage.

**Table 3 jcm-10-04382-t003:** Clinical outcomes in morbidly obese compared to non-morbidly obese.

BMI	Normal/Overweight	Obese (Obesity Class 1 and 2)	Morbidly Obese (Obesity Class III)	*p* Value
Number of subjects	*n* = 11,028 (66.5)	*n* = 4547 (27.4)	*n* = 1004 (6.1)	
Outcomes				
*LOS days median (IQR)	5 (2, 11)	5 (2, 10)	5 (2, 12)	0.030
LOS *n* (%)				
≤2 days 3–14 days 15–28 days >28 days	3276 (29.7) 5801 (52.6) 1185 (10.8) 766 (6.9)	1266 (27.8) 2511 (55.2) 510 (11.3) 260 (5.7)	259 (25.8) 556 (55.4) 119 (11.9) 70 (6.9)	0.002
ICU admission rate	1041 (9.4)	400 (8.8)	98 (9.8)	0.393
ICU hours mean (SD)	14.3 (103.2)	10.2 (60.2)	11.9 (68.4)	0.051
In hospital mortality	95 (0.90)	41 (0.93)	5 (0.51)	0.205
No of complications mean (SD)	0.9 (2.1)	0.8 (2.0)	0.9 (1.9)	0.528
Readmissions within 7 days	531 (4.8)	196 (4.3)	53 (5.3)	0.271
Readmissions within 30 days	1332 (12.1)	538 (11.8)	127 (12.7)	0.759

*LOS adjusted for mortality. LOS, length of hospital stay; IQR, interquartile range; ICU, intensive care unit.

**Table 4 jcm-10-04382-t004:** Multilevel mixed effects regression models comparing clinical outcomes in morbidly obese with normal/overweight patients.

Outcome	OR/IRR	95% CI	*p* Value
LOS			
Unadjusted model	1.04 *	1.02–1.06	<0.001
Adjusted model excluding Charlson index	1.05 *	1.03–1.07	<0.001
Adjusted model with Charlson index	1.04 *	1.02–1.07	<0.001
In-hospital mortality			
Unadjusted model	0.57	0.23–1.41	0.224
Adjusted model excluding Charlson index	0.67	0.27–1.67	0.392
Adjusted model with Charlson index	0.65	0.26–1.62	0.354
ICU admission			
Unadjusted model	1.06	0.86–1.32	0.739
Adjusted model excluding Charlson index	1.08	0.87–1.35	0.493
Adjusted model with Charlson index	1.06	0.85–1.32	0.584
ICU LOS			
Unadjusted model	1.04 *	0.83–1.30	0.058
Adjusted model excluding Charlson index	1.05 *	0.84–1.31	0.675
Adjusted model with Charlson index	1.03 *	0.82–1.30	0.801
Complications			
Unadjusted model	1.00	0.94–1.08	0.793
Adjusted model excluding Charlson index	1.02	0.95–1.09	0.587
Adjusted model with Charlson index	1.00	0.93–1.07	0.996
30 day readmission			
Unadjusted model	1.06	0.87–1.29	0.571
Adjusted model excluding Charlson index	1.04	0.86–1.27	0.674
Adjusted model with Charlson index	1.02	0.83–1.25	0.818

* IRR, incident risk ratio; OR, odds ratio; LOS, length of hospital stay; ICU, intensive care unit.

**Table 5 jcm-10-04382-t005:** Multilevel mixed effects regression models comparing clinical outcomes in morbidly obese with obese patients.

Outcome	OR/IRR	95% CI	*p* Value
LOS			
Unadjusted model	1.09 *	1.07–1.12	<0.001
Adjusted model excluding Charlson index	1.14 *	1.11–1.16	<0.001
Adjusted model with Charlson index	1.13 *	1.11–1.16	<0.001
In-hospital mortality			
Unadjusted model	0.55	0.22–1.40	0.208
Adjusted model excluding Charlson index	0.67	0.26–1.70	0.395
Adjusted model with Charlson index	0.66	0.26–1.62	0.354
ICU admission			
Unadjusted model	1.12	0.96–1.22	0.208
Adjusted model excluding Charlson index	1.16	0.91–1.47	0.226
Adjusted model with Charlson index	1.15	0.91–1.46	0.251
ICU LOS			
Unadjusted model	1.12 *	0.89–1.41	0.334
Adjusted model excluding Charlson index	1.16 *	0.91–1.47	0.226
Adjusted model with Charlson index	1.15 *	0.91–1.46	0.251
Complications			
Unadjusted model	1.05	1.01–1.09	0.010
Adjusted model excluding Charlson index	1.10	1.02–1.19	0.012
Adjusted model with Charlson index	1.09	1.02–1.18	0.017
30 day readmission			
Unadjusted model	1.08	0.88–1.34	0.437
Adjusted model excluding Charlson index	1.09	0.88–1.34	0.416
Adjusted model with Charlson index	1.08	0.88–1.33	0.465

* IRR, incident risk ratio; OR, odds ratio; LOS, length of hospital stay; ICU, intensive care unit.

## Data Availability

The data presented in this study are available on request from the corresponding author only after permission is granted by the ethics committee.

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
