# Peer review of "The Impact of Morbid Obesity on the Health Outcomes of Hospital Inpatients: An Observational Study"

_jcm, 2021, doi:10.3390/jcm10194382_

Round 1

Reviewer 1 Report

In this manuscript entitled “The Impact of Morbid Obesity on the Health Outcomes of Hospital Inpatients: an observational study” Fusco and her co-workers investigated retrospectively nutritional data of hospitalised patients aged 18-79 years, with a BMI ≥18.5 kg/m2 admitted over a period of 4 years. They conclude that morbid obese had longer hospital stays compared to non-morbid obese. The study design is good and number of study subjects 28900. They have collected the data retrospectively from the malnutrition screening database used by two tertiary-level hospitals.

The research question is highly important and warrants further investigation. The paper is generally well written and structured. However, in my opinion the paper has some shortcomings in regards to study subject selection, and I feel this dataset has not been utilized to its full extent.

Difference in hospital stay between the two groups was minor (5 (2,11) vs. 5 (2,10), P-value=0.006). Did the authors compare length of hospital stay between sub-groups of normal weight, class 1-3 obese?

The Malnutrition Universal screening tool dataset contains only 10% of patient admitted. How was the selection made? Which patients were selected for screening tool?

How is the patient selection made between secondary- and tertiary-level centers in Australia? 

Author Response

In this manuscript entitled “The Impact of Morbid Obesity on the Health Outcomes of Hospital Inpatients: an observational study” Fusco and her co-workers investigated retrospectively nutritional data of hospitalised patients aged 18-79 years, with a BMI ≥18.5 kg/m2 admitted over a period of 4 years. They conclude that morbid obese had longer hospital stays compared to non-morbid obese. The study design is good and number of study subjects 28900. They have collected the data retrospectively from the malnutrition screening database used by two tertiary-level hospitals.

The research question is highly important and warrants further investigation. The paper is generally well written and structured. However, in my opinion the paper has some shortcomings in regards to study subject selection, and I feel this dataset has not been utilized to its full extent.

Difference in hospital stay between the two groups was minor (5 (2,11) vs. 5 (2,10), P-value=0.006). Did the authors compare length of hospital stay between sub-groups of normal weight, class 1-3 obese?

Response: We have now divided the data set as per the advice from the reviewers into 3 categories: normal/overweight (BMI 18.5-29.9), Obese class 1-2 (BMI 30-39.9) and Morbidly obese (BMI>40) and compared clinical outcomes.

When compared to patients with a normal BMI/overweight and those with obesity class 1 and 2, LOS was significantly higher among morbidly obese patients after adjusted multivariable mixed effect regression analysis. However, other clinical outcomes were not significantly different between 3 groups. We have now modified the abstract, methods section, results and discussion along with tables and have also added a new table (Table 5), to show comparisons between the 3 groups.

Please refer to page 1 (Abstract section), pages 2-3 (methods section), pages 3-9 (results section) and pages 8 and 11 (discussion section).

The Malnutrition Universal screening tool dataset contains only 10% of patient admitted. How was the selection made? Which patients were selected for screening tool?

Response: In our hospitals, it is a policy that every patient who needs admission gets MUST screening. However, MUST screening rates are poor and our previous studies 1, 2 have suggested that a range of factors may be responsible for this poor practice. Because not all patients had MUST screening data available, so we acknowledge that there is a selection bias and thus a major limitation of this study, which has been documented in the limitations section on page 20. The BMI distribution in our study, however, closely matches another Australian study3 which captured data of hospitalised patients.

“The dataset contains only about 10% of patients admitted to the two institutions over the period of study which could have introduced a selection bias in this study. This poor capture of patient anthropometry is disappointing but not unusual. However, the BMI distribution in our study matches another Australian study 3, which captured data of all hospitalised patients.”

How is the patient selection made between secondary- and tertiary-level centers in Australia? 

Response:

Patients who requiring a higher level of care in hospital such as highly specialised treatment e.g., cardiac surgery, cancer treatment etc. are admitted in tertiary settings after referral from primary care physicians or after triage for admission requiring acute care. Other patients who require acute hospitalisation or investigations may be admitted in secondary hospitals. In general, patients who are anticipated to need a subspeciality input by the ambulance personnel or GPs are preferably referred for admission under tertiary hospitals. However, there exists a potential overlap and secondary hospitals may end up seeing a number of such patients if EDs are busy to share care burden.

References

  1. Sharma Y, Miller M, Kaambwa B, Shahi R, Hakendorf P, Horwood C, Thompson C. Malnutrition and its association with readmission and death within 7 days and 8–180 days postdischarge in older patients: a prospective observational study. BMJ open. 2017;7. doi: 10.1136/bmjopen-2017-018443
  2. Sharma Y, Miller M, Kaambwa B, Shahi R, Hakendorf P, Horwood C, Thompson C. Factors influencing early and late readmissions in Australian hospitalised patients and investigating role of admission nutrition status as a predictor of hospital readmissions: a cohort study. BMJ open. 2018;8:e022246. doi: 10.1136/bmjopen-2018-022246
  3. Dennis DM, Carter V, Trevenen M, Tyler J, Perrella L, Lori E, Cooper I. Do acute hospitalised patients in Australia have a different body mass index to the general Australian population: a point prevalence study? Aust Health Rev. 2018;42:121-9. doi: 10.1071/ah16171

Reviewer 2 Report

Comments to the Author

Thanks to the authors to letting me review the manuscript. The aim of this retrospective study is to establish whether morbid obesity leads to a worsening of the outcome in hospitalized patients. This is of interest, especially in light of the fact that there are conflicting results in the scientific literature.

The manuscript has merit, but there are also some critical issues.

1) From a methodological point of view, the statistical analysis is well conducted. What puzzles me is the choice of having rigidly dichotomized patients on the basis of the presence or not of morbid obesity (BMI >40). The sample size would make it possibile to conduct a statistical analysis by dividing the population into 3 groups: normal weight, obese (1-2 class, 30 < BMI ≥ 40) and morbidly obese, checking whether there are any differences between mild or morbid obesity. In fact, studies have recently been conducted that indicate that all classes of obesity, and not only the morbid obesity, are associated with a prolonged hospitalization, i.e 1) NT Nguyen, doi: 10.1371/journal.pone.0154599 - 2) Moriconi et al, doi: 10.1016/j.orcp.2020.05.009.

2) page 3 row 131, the number of non morbidly obese groups is wrong (1575), please correct.

Author Response

Thanks to the authors to letting me review the manuscript. The aim of this retrospective study is to establish whether morbid obesity leads to a worsening of the outcome in hospitalized patients. This is of interest, especially in light of the fact that there are conflicting results in the scientific literature.

The manuscript has merit, but there are also some critical issues.

  • From a methodological point of view, the statistical analysis is well conducted. What puzzles me is the choice of having rigidly dichotomized patients on the basis of the presence or not of morbid obesity (BMI >40). The sample size would make it possibile to conduct a statistical analysis by dividing the population into 3 groups: normal weight, obese (1-2 class, 30 < BMI ≥ 40) and morbidly obese, checking whether there are any differences between mild or morbid obesity. In fact, studies have recently been conducted that indicate that all classes of obesity, and not only the morbid obesity, are associated with a prolonged hospitalization, e 1) NT Nguyen, doi: 10.1371/journal.pone.0154599 - 2) Moriconi et al, doi: 10.1016/j.orcp.2020.05.009.

Response: We have now divided the data set as per the advice from the reviewers into 3 categories: normal/overweight (BMI 18.5-29.9), Obese class 1-2 (BMI 30-39.9) and Morbidly obese (BMI>40) and compared clinical outcomes.

When compared to patients with a normal/overweight BMI and those with obesity class 1 and 2, LOS was significantly higher among morbidly obese patients after adjusted multivariable mixed effect regression analysis. However, other clinical outcomes were not significantly different between 3 groups. We have now modified the abstract, methods section, results and discussion along with tables and have also added a new table (Table 5), to show comparisons between the 3 groups.

Please refer to page 3 Abstract section, page 6 methods section, pages 7-16 results section and pages 16-17 discussion section.

  • page 3 row 131, the number of non morbidly obese groups is wrong (1575), please correct.

Response: This has now been corrected.

“For this study, we compared the clinical characteristics and outcomes between 3 groups: normal/overweight patients n=11,028 (66.5%), obese patients (obesity class 1 and 2) n=4547 (27.4%) and the morbidly obese group n=1004 (6.1%).”

References

  1. Sharma Y, Miller M, Kaambwa B, Shahi R, Hakendorf P, Horwood C, Thompson C. Malnutrition and its association with readmission and death within 7 days and 8–180 days postdischarge in older patients: a prospective observational study. BMJ open. 2017;7. doi: 10.1136/bmjopen-2017-018443
  2. Sharma Y, Miller M, Kaambwa B, Shahi R, Hakendorf P, Horwood C, Thompson C. Factors influencing early and late readmissions in Australian hospitalised patients and investigating role of admission nutrition status as a predictor of hospital readmissions: a cohort study. BMJ open. 2018;8:e022246. doi: 10.1136/bmjopen-2018-022246
  3. Dennis DM, Carter V, Trevenen M, Tyler J, Perrella L, Lori E, Cooper I. Do acute hospitalised patients in Australia have a different body mass index to the general Australian population: a point prevalence study? Aust Health Rev. 2018;42:121-9. doi: 10.1071/ah16171

Round 2

Reviewer 1 Report

The authors have responded to the concerns raised in my review by appropriately altering the interpretation of the results, including new data, and expanding the limitations section.

Reviewer 2 Report

The authors have answered all of my questions and the additional analysis improved the manuscript. The paper can be accepted for publication.